# Effects of Biased Analogues of the Kappa Opioid Receptor Agonist, U50,488, in Preclinical Models of Pain and Side Effects

**DOI:** 10.3390/molecules30030604

**Published:** 2025-01-29

**Authors:** Ross van de Wetering, Loan Y. Vu, Lindsay D. Kornberger, Dan Luo, Brittany Scouller, Sheein Hong, Kelly Paton, Thomas E. Prisinzano, Bronwyn M. Kivell

**Affiliations:** 1School of Biological Sciences, Centre for Biodiscovery, Victoria University of Wellington, Wellington 6012, New Zealandscoullerbrittany@gmail.com (B.S.);; 2Department of Pharmaceutical Sciences, University of Kentucky, Lexington, KY 40506, USA; loan.vu@uky.edu (L.Y.V.); lindsay.kornberger17@gmail.com (L.D.K.); dan.luo@uky.edu (D.L.);

**Keywords:** kappa opioid receptor, anti-nociception, pain, side effects, biased agonism, signaling bias, functional selectivity, U50,488

## Abstract

Kappa opioid receptor (KOR) agonists have well-established antinociceptive effects. However, many KOR agonists have negative side effects, which limit their therapeutic potential. Some researchers have suggested that the development of biased agonists that preferentially stimulate KOR G-protein pathways over β-arrestin pathways may yield drugs with fewer adverse side effects. This was investigated in the current study. We describe the synthesis and characterization of three U50,488 analogues, **1**, **2**, and **3**. We evaluated the acute and chronic antinociceptive effects of these compounds in mice using the warm-water tail flick assay and in a paclitaxel-induced neuropathic pain model. Side effects were investigated using open-field, passive wire hang, rotarod, elevated zero maze, conditioned place aversion, and whole-body plethysmography, with some tests being conducted in KOR or β-arrestin2 knock out mice. All compounds were highly potent, full agonists of the KOR, with varying signaling biases in vitro. In the warm-water tail withdrawal assay, these agonists were ~10 times more potent than U50,488, but not more efficacious. All KOR agonists reversed paclitaxel-induced neuropathic pain, without tolerance. Compound **3** showed no significant side effects on any test. Signaling bias did not correlate with the antinociceptive or side effects of any compounds and knockout of β-arrestin2 had no effect on U50,488-induced sedation or motor incoordination. These findings highlight the therapeutic potential of **3**, with its lack of side effects typically associated with KOR agonists, and also suggest that G-protein signaling bias is a poor predictor of KOR agonist-induced side effects.

## 1. Introduction

The kappa opioid receptor (KOR) is an inhibitory G-protein-coupled receptor activated by the endogenous peptide, dynorphin, which is widely distributed throughout the central nervous system and in some peripheral tissues [1,2,3]. As a pharmacological target, the KOR has recently gained attention for its role in modulating myelination and neuroinflammation [4,5,6,7,8], with KOR agonists offering potential as a first-in-class remyelination treatment for demyelinating diseases such as multiple sclerosis. KOR agonists also exhibit anti-pruritic (anti-itch) effects [9], and two are now approved for the treatment of pruritus in hemodialysis patients: Nalfurafine (Remitch) is a centrally active KOR agonist approved in Japan since 2009 [10] while difelikefalin (KORSUVA) is a peripherally restricted agonist approved by the U.S. Food and Drug Administration in 2021 [11]. In the current study, we focus on the well-established anti-nociceptive effects of KOR agonists [12,13,14,15,16,17].

KOR agonists have demonstrated anti-nociceptive efficacy in various preclinical models of acute mechanical, thermal, and chemical pain, as well as in inflammatory, neuropathic, and cancer-related pain [18,19,20,21,22,23,24,25]. As analgesics, KOR agonists could offer several advantages over mu opioid receptor (MOR) agonists (e.g., morphine, fentanyl, oxycodone). Notably, KOR agonists are non-addictive [26,27], do not induce respiratory depression [28,29] or gastrointestinal side effects [30], and show less antinociceptive tolerance [19]. These characteristics underscore the potential of KOR agonists as a safer alternative to MOR agonists for pain management.

However, KOR agonists have their own adverse side effects, including sedation, anxiety, and dysphoria/aversion [31,32,33,34,35,36,37,38,39,40,41], which has limited their clinical development. In an effort to minimize these side effects, many researchers have sought to develop novel compounds that preferentially stimulate KOR G-protein signaling over β-arrestin [14,15,16,17,18]. This idea, at least within the KOR field, largely stems from research showing that the blockading of p38 mitogen-activated protein kinase (MAPK) activation using the inhibitor, SB203580, or a conditional knockout of p38α MAPK, prevented conditioned place aversion caused by the prototypical KOR agonist, U50,488, in mice [32,42]. Activation of p38 MAPK is thought to be dependent on β-arrestin2 recruitment by KOR [43]. Therefore, it was hypothesized that reduced β-arrestin2 recruitment, which does not appear to impact the antinociceptive effects of KOR agonists [44,45,46], may increase the therapeutic windows [47].

While several novel G-protein-biased KOR agonists have shown reduced side effects compared to typical KOR agonists [14,15,16,17,18], it is not clear whether the G-protein bias of these compounds is indeed the cause for their reduced side effects since some very highly G-protein biased KOR agonists still showed significant aversive or sedative effects [48,49]. Moreover, subsequent research has shown that knockout of β-arrestin2 had no impact on conditioned place aversion or locomotor sedation caused by a variety of KOR agonists, but it did attenuate KOR-induced motor incoordination on the rotarod [49]. These inconsistent findings question whether G-protein bias is a useful metric to pursue for KOR drug development.

To better understand the pharmacological properties associated with therapeutically viable KOR agonists, we prepared the novel compounds, **1**, and **2** (Figure 1). Previously, Weerawarna and colleagues found that (S)-*N*-methyl-2-phenyl-*N*-(1-phenyl-2-(pyrrolidin-1-yl)ethyl)acetamide (compound **A**; Figure 1) had high affinity for the KOR [50]. IC-199,441 (3,4-dichlorophenylacetamide analog) also had selective and high affinity for the KOR [51]. It was envisioned that modifying phenylacetamide would result in highly potent compounds with the potential for reduced side effects. Previous structure–activity relationship studies suggested that the insertion of an oxygen atom would be well tolerated. It was also envisioned that modification of the halogens on the phenyl ring might offer enhanced metabolic stability and perhaps an improvement in side effects. To further understand the structure–activity relationship of this chemotype at KORs, we also prepared the pyridine analogue, **3** (Figure 1) [52]. This bioisosteric replacement suggests a molecule with a lower log D (1-octanol-water coefficient at various pH values) and is predicted to increase solubility and potency as well as decrease potential promiscuity and side effects [53,54,55]. We then determined the signaling bias of these three U50,488-like compounds and evaluated their effects in various preclinical models of pain and side effects, including studies conducted in β-arrestin2 knockout mice.

## 2. Results

### 2.1. Synthesis

Synthesis of the phenylacetamides, **1** and **2**, is shown in Figure 1. Commercially available (S)-N-methylphenyl pyrrolidine **4** was coupled with 2-(3,4-dichlorophenoxy) acetic acid to afford **1** and 2-(2,6-dichlorophenyl) acetic acid to afford **2**. The synthesis of **3** is also shown in Figure 1. Commercially available pyrrolidine was used to ring open 3-(oxiran-2-yl)pyridine **5** to afford **6**. Transformation of alcohol into mesylate and in situ nucleophilic substitution with methyl amine gave **7** [56]. Amide coupling with 2-(3,4-dichlorophenyl)acetic acid yielded **3**. Full details on the synthesis and purification of these compounds are described in the Appendix A.

### 2.2. Functional Activity and Signaling Bias

G-protein activity (inhibition of forskolin-induced cAMP) and β-arrestin2 recruitment was determined using the HitHunter^®^ and PathHunter^®^ detection assays (DiscoverX^®^, Eurofin, Fremont, CA, USA) (Table 1). Compounds **1**, **2**, and **3** were all more potent than U50,488 in both assays. Calculation of signaling bias using their relative activities indicated that **2** and **3** were more biased for G-protein signaling relative to U50,488, while **1** showed a similar signaling bias to U50,488 (Table 1).

### 2.3. Pain Models

We then evaluated these three compounds in various preclinical pain models. First, we collected cumulative dose–response data using the warm water tail-withdrawal assay (Figure 2a). All compounds were significantly more potent (logED_50_) but showed similar efficacy (E_max_) relative to U50,488. At equipotent ED_50_ doses, all drugs produced a similar time course in the tail withdrawal assay, with peak effects between 15 and 30 min (Figure 2b). Repeating this experiment in KOR knockout mice completely abolished the effects of all compounds, confirming that the antinociceptive effects of these compounds are solely mediated by the KOR (Figure 2c).

Next, we evaluated these compounds in the paclitaxel-induced neuropathy model of chronic pain (Figure 3). Paclitaxel administration rapidly induced both mechanical (Figure 3a) and thermal (Figure 3b) allodynia, as observed during the 15-day induction phase. Animals subsequently treated with vehicle remained significantly more sensitive to both mechanical and thermal stimuli for the remainder of the 42-day study relative to healthy (non-paclitaxel-treated) controls. During the treatment phase, all KOR compounds significantly attenuated both mechanical (Figure 3a) and thermal (Figure 3b) allodynia, with responses equivalent to healthy controls. Importantly, no significant tolerance was observed following repeated daily treatment of any KOR agonist in these tests.

### 2.4. Side Effects

We initially investigated the sedative-like side effects in the open-field locomotor activity test (Figure 4), using ~ED_50_ doses calculated from the tail withdrawal assay. U50,488, **1**, and **2** significantly decreased distance traveled over time, indicative of sedation (Figure 4a). No effect of these drugs was observed in KOR knockout mice, however, confirming KOR dependency (Figure 4b). Interestingly, **3**, showed no significant sedative effects at this dose (Figure 4a). To confirm this finding, and to investigate the role of β-arrestin2 signaling, we repeated this experiment with U50,488 and with higher doses of **3** in both wild-type and β-arrestin2 knockout mice. Knockout of β-arrestin2 had no impact of U50,488-induced sedation, while **3** had no significant sedative effect at any dose (Figure 4c,d). When tested with a 15 min pre-treatment period to evaluate novelty/anxiety-induced exploration, U50,488 significantly decreased the number of entries into the central zone, suggesting an anxiogenic-like effect (Figure 4f), though this could also be due to the sedative effects of the compound, since there was no effect on the percentage of time spent in the center (Figure 4g). In contrast, **3** continued to show no sedative effects, even at 2 × ED_50_ doses (Figure 4e), and also had no effect on the number of entries (Figure 4f) or the percentage of time spent (Figure 4g) in the center of the open-field arena, indicating no impact on anxiety-like thigmotaxis behavior.

To further evaluate sedation, grip strength, and motor coordination, we conducted passive wire hang and rotarod tests (Figure 5). In the passive wire hang test, U50,488, **1**, and **2** significantly decreased hang latency compared to vehicle-treated controls, indicative of increased fatigue/reduce grip strength, whereas **3** had no significant effect (Figure 5a). Repeating this study with higher doses of **3** produced the same result (Figure 5b). On the rotarod test, U50,488 significantly decreased motor coordination whereas **3** had no effect at either dose tested (Figure 5c). Furthermore, knockout of β-arrestin2 had no impact on the effect of U50,488 in the rotarod test (Figure 5d).

Next, we used the elevated zero maze to more thoroughly investigate anxiety-like behavior, and used the conditioned place aversion assay to evaluate aversion (Figure 6). In the elevated zero maze, the results mirrored those from the open-field test; U50,488 produced a significant reduction in the total distance traveled (Figure 6a) as well as the number of entries into the open arms of the elevated zero maze (Figure 6b), while **3** had no effect on either measure at any dose. In the conditioned placed aversion test, **3** had no effect at any dose tested, but there was a significant decrease in the % preference for the U50,488-paired chamber, indicative of aversion (Figure 6d).

Lastly, we carried out whole-body plethysmography to determine the effect of our KOR agonists on respiration (Figure 7). Unlike MOR agonists, KOR agonists are not known to induce respiratory depression [28,29]. Accordingly, we found that U50,488, **1**, and **2** had no significant effect on respiratory frequency (Figure 7a,b) or tidal volume (Figure 7c,d) in wild-type or KOR knockout mice, though we must acknowledge the low sample size of wild-type mice used in this experiment may limit statistical power. Due to a limited supply, we were not able to test **3** here.

## 3. Discussion

This study aimed to evaluate the therapeutic potential of three compounds, derived from the KOR agonist, U50,488, and investigate the role of G-protein signaling bias. We identified one compound, **3,** as a therapeutically viable option, with its remarkable lack of side effects. Interestingly, the signaling bias of **1** (balanced), **2** (G-protein), and **3** (G-protein) had no correlation with their antinociceptive effects or their side effects. Moreover, our tests in β-arrestin2 knockout mice suggest that β-arrestin2 had no role in mediating KOR-induced sedation or motor incoordination.

### 3.1. Therapeutic Potential of **1**, **2**, and **3**

All tested KOR agonists had significant effects in the warm water tail-withdrawal assay, an acute thermal nociception model. While these compounds were all ~10× more potent than the prototypical KOR agonist, U50,488, they were not more efficacious, and the maximum efficacy (~50% MPE) was notably lower than that achieved with MOR agonists (100% MPE) [57]. This may limit their therapeutic potential for managing severe acute pain. However, KOR agonists have been shown to be more effective at attenuating acute mechanical and chemical nociception compared to thermal models of nociception [58].

Where KOR agonists may excel is in the management of chronic pain. Paclitaxel, like many other chemotherapy drugs, causes chronic neuropathic pain in a large proportion of patients [59]. Combined with other cancer-related pains, this leads to a strong demand for safe and effective chronic pain relief. MOR agonists, while highly effective for acute severe pain, have significant on-target side effects that limit their safety and utility in managing such chronic pain [60]. Our results suggest that KOR agonists may provide a safer alternative. We found that all KOR agonists completely reversed paclitaxel-induced allodynia in mice. Importantly, unlike MOR agonists [19], we observed no evidence of tolerance, even after 3 weeks of daily treatment. Paclitaxel causes microtubule dysfunction and impedes axonal transport, leading to peripheral nerve fiber damage, demyelination, and inflammation, as well as a dysregulation of Ca^2+^ signaling and altered circuitry throughout the peripheral and central nervous system [61,62]. Therefore, the effectiveness of KOR agonists in this model may be due to their antinociceptive effects in combination with their remyelinating and anti-inflammatory effects, both centrally and peripherally [13].

While KOR agonists do not have the same severe side effects as MOR agonists, other side effects including sedation, anxiety, and dysphoria/aversion have limited their clinical development [31,32,33,34,35,36,37,38,39,40,41]. Indeed, as expected, we found that U50,488 showed significant sedative effects, significant reductions in grip strength and motor coordination, potential anxiety-like effects, and significant aversion. The novel compounds, **1** and **2** also showed similar significant sedative-like effects. In contrast, **3** showed no significant side effects in any test conducted, many of which were replicated 2–3 times with different experimenters and with doses up to 2× the antinociceptive ED_50_. This remarkable lack of side effects compared to other KOR agonists [12,13,14,15,16], highlights the need for further detailed investigations into the therapeutic applications of **3**, in managing chronic pain, as an anti-pruritic, or as a remyelinating agent.

One possible explanation for the unique side effect profile of **3** could be due to differences in the LogD value (**3**: LogD = 1.9, **1**: LogD = 2.7, **2**: LogD = 2.6). The replacement of a phenyl ring to a pyridine **3** results in a less lipophilic compound which could impact the ligand–protein affinity [63]. Daibani and colleagues reported in their molecular dynamic simulation that U50,488 inserted a hydrophobic group, which was the cyclohexane ring in the central pocket [64]. If our analogues possessed a similar binding mode, the interaction of pyridine **3** to the central pocket would result in a unique binding mode compared to phenyl ring present in **1** and **2**. How this might impact downstream cellular signaling and side effects is not entirely clear, but as discussed below, we do not think β-arrestin2 signaling plays an important role.

### 3.2. G-Protein Bias and Role of β-Arrestin2 Signaling

In the search for therapeutically viable KOR agonists with fewer adverse side effects, many research groups, including ours, have aimed to develop compounds with G-protein signaling bias. Although several novel G-protein biased KOR agonists have shown improved side effect profiles compared to typical KOR agonists [14,15,16,17,18], it is difficult to attribute any reduction in side effects to their G-protein bias for two primary reasons.

Firstly, these findings are largely correlational, and do not imply a causal role of G-protein bias in minimizing adverse side effects. Moreover, there are also several contradictory findings, with some highly G-protein-biased compounds still showing significant side effects. For example, the salvinorin A derivative, RB-64, caused significant conditioned place aversion, similar to salvinorin A at equivalent antinociceptive doses, despite being 96 times more G-protein-biased compared to salvinorin A [49]. In another example with diphenethylamine derivatives, HS665 (with a G-protein bias factor of 389 compared to U69,539) showed significant conditioned place aversion, while HS666 (with a G-protein bias factor of 62 compared to U69,539) did not [48]. In our study, **1** (bias factor 0.9) and **2** (bias factor 2.2) showed significant sedative effects, while **3** (bias factor 1.8) did not. To reliably determine if G-protein bias correlates with greater therapeutic windows, a systematic investigation using several KOR agonists with different G-protein bias is required, as has been explored with some MOR agonists [57]. Furthermore, it would be beneficial to test β-arrestin-biased compounds to determine whether the opposite holds true, i.e., whether β-arrestin-biased KOR agonists show more side effects.

Second, studies that have directly investigated the causal role of G-protein bias/β-arrestin2 signaling have shown conflicting results. Inhibition of β-arrestin2-dependant p38 MAPK signaling prevented conditioned place aversion to U50,488, suggesting a role of β-arrestin2 and p38 MAPK in KOR-induced aversion [32,42]. In contrast, β-arrestin2 knockout mice still showed significant conditioned place aversion to the KOR agonists, U69,593, salvinorin A, and RB-64, with no difference compared to wild-type mice [49]. The authors suggested that KOR-induced aversion could instead be mediated by G-protein signaling, by p38-MAPK-independent mechanisms, or that p38-MAPK could be activated by β-arrestin2-independent mechanisms [49]. Given the global knockout model used in their study, as well as in the current study, it is possible that β-arrestin1 may compensate as the primary mediator of p38 MAPK signaling. In regard to sedation, White and colleagues found that knockout of β-arrestin2 had no impact on locomotor sedation caused by U69,593 or salvinorin A [49], which is in line with the findings of the current study. They also found that both U69,593 and salvinorin A still caused significant motor incoordination on the rotarod in β-arrestin2 knockout mice, though this was attenuated compared to wild-type controls [49]. These results suggest that β-arrestin2 signaling may contribute to motor incoordination, but not necessarily. In the current study, we found that knockout of β-arrestin2 had no effect on U50,488-induced motor incoordination. The reason for this discrepancy is unclear, but it may be a drug-dependent effect.

Taken together, current evidence does not support a clear role of β-arrestin signaling in KOR agonist-induced side effects, and G-protein bias does not appear to be a reliable predictor of therapeutic viability. One important caveat that is important to also note regards the limitations in quantifying signaling bias. Recent reviews have discussed how the use of different cell lines, receptor species, assays, reference ligands, and models to determine signaling bias can have a drastic impact on the results, which makes it difficult to consistently compare findings across studies [12,14,16,17,65,66]. Moreover, the relevance of signaling bias in immortalized cell-cultures, often with excessive receptor expression confounding accurate efficacy measurements [67], has also been questioned, with many laboratories looking to determine signaling bias in primary cultures [68,69,70]. As has been suggested in the MOR field, it may be that low intrinsic efficacy or partial agonism may better correlate with a reduction in certain side effects, more so than G-protein bias per se [67].

## 4. Materials and Methods

### 4.1. Drugs

**1**, **2**, and **3** were synthesized and tested for purity (>95%) using high-performance liquid chromatography (HPLC) and nuclear magnetic resonance (NMR). Full methods for the synthesis and purification as well as HPLC and NMR data for these compounds can be found in the Appendix A.

U50,488, U69,593, and salvinorin A were kindly provided by the National Institute on Drug Abuse Drug Supply Program (Rockville, MD, USA). Paclitaxel (6 mg/mL in 0.9% saline) was purchased from Hospira, Wellington, New Zealand. For the cumulative dose effect tail-withdrawal experiment, KOR agonists were dissolved in a 2:1:7 ratio of DMSO, Tween-80, and 0.9% saline and administered at a volume of 5 µL/g subcutaneously (s.c.), or, for all other pain and side effect experiments, at 10 µL/g i.p. Paclitaxel was further diluted in ethanol, kolliphor, and 0.9% saline to give a final ratio of 1:1:18 and was administered at a volume of 10 µL/g i.p.

### 4.2. Cell Lines and Cell Culture

Both the cAMP HitHunter^®^ CHO-K1 OPRK1 Gi Cell Line (catalog # 95-0088C2) and PathHunter^®^ U2OS OPRK1 β-Arrestin Cell Lines (catalog # 93-0234C3) were purchased from Eurofins DiscoverX (Fremont, CA, USA). The cAMP HitHunter^®^ cell line was maintained in F-12 media supplemented with 10% fetal bovine serum (Life Technologies, Grand Island, NY, USA), 1% penicillin/streptomycin/l-glutamine (Life Technologies), and 800 µg/mL Geneticin (Mirus Bio, Madison, WI, USA). The PathHunter^®^ U2OS cell line was maintained in MEM media supplemented with 10% fetal bovine serum, 1% penicillin/streptomycin/l-glutamine, 500 µg/mL Geneticin, and 250 µg/mL Hygromycin B (Mirus Bio, Madison, WI, USA). All cells were grown at 37 °C and 5% CO_2_ in a humidified incubator.

### 4.3. Forskolin-Induced cAMP Accumulation

cAMP HitHunter^®^ cells were seeded (10,000 cells/well) to 384-well white tissue culture plates and incubated at 37 °C overnight. The cells were treated with 9-doses of test compounds in the presence of forskolin for 30 min at 37 °C followed by detection using the HitHunter^®^ cAMP assay for small molecules assay kit (Eurofins DiscoverX) according to the manufacturer’s directions. BioTek Synergy H1 hybrid reader, BioTek Cytation 5 reader, and Gen5 software version 3.11 (BioTek, Winooski, VT, USA) were used to quantify the luminescence generated. Data were blank-subtracted with vehicle control, normalized to forskolin controls, and analyzed with non-linear regression by GraphPad Prism version 8 (GraphPad, La Jolla, CA, USA).

### 4.4. β-Arrestin2 Recruitment Assay

PathHunter^®^ cells were seeded (5000 cells/well) into 384-well white tissue culture plates and incubated at 37 °C overnight. The cells were treated with 9–10 doses of test compounds for 30 min at 37 °C followed by the detection using the PathHunter^®^ detection kit (Eurofins DiscoverX) according to the manufacturer’s directions. BioTek Synergy H1 hybrid reader, BioTek Cytation 5 reader, and Gen5 software (BioTek, Winooski, VT, USA) were used to quantify the luminescence generated. Data were blank-subtracted with vehicle control, normalized to the reference compound U50,488H, and analyzed with non-linear regression with GraphPad Prism version 8.

### 4.5. Bias Calculation

The following formula, with U50,488 as the control ligand, was used to calculate the bias factor [71,72]:$$\log\left(bias\;factor\right)={\log\left(\frac{E_{\max\left(test\right)}\times {EC}_{50\left(control\right)}}{{EC}_{50\left(test\right)}\times E_{\max\left(control\right)}}\right)}_{Gprotein}-{\log\left(\frac{E_{\max\left(test\right)}\times {EC}_{50\left(control\right)}}{{EC}_{50\left(test\right)}\times E_{\max\left(control\right)}}\right)}_{\beta -arrestin2}$$

Using this formula, a bias factor of 1 is a balanced agonist, more than 1 is a G-protein-biased agonist, and less than 1 is a β-arrestin2-biased agonist, relative to U50,488.

### 4.6. Subjects

Adult female and male (whole body plethysmography) or male (all other experiments) C57BL/6J, homozygous KOR knockout (B6.129S2-Oprk1tm1Kff/J, stock # 007558. The Jackson Laboratory, Bar Harbor, ME, USA), and homozygous β-arrestin2 knockout (B6.129X1(Cg)-Arrb2tmRjl/J, stock # 023852, The Jackson Laboratory) mice aged 10–12 weeks were used. Mice were bred and housed (2–5/cage) in the Victoria University of Wellington Small Animal Facility. The facility was temperature (19–21 °C) and humidity (55%) controlled, with a 12 h light cycle (lights on at 07:00). Food and water were available ad libitum. For all behavioral experiments, mice were habituated to the experimental environment and experimenter for 30 min per day, for at least 4 days prior to testing. All experimental protocols were approved by the Victoria University of Wellington Animal Ethics Committee.

### 4.7. Warm Water Tail-Withdrawal

The tail-withdrawal assay was carried out as previously described [18]. Briefly, mice were allowed to habituate to plexiglass restraints (internal diameter 24 mm) for 5 min per day for 4 days prior to testing. On the test day, the distal 1/3 of the tail was then immersed in warm water (50 ± 0.5 °C) and the time taken to withdraw the tail was recorded, with a maximum latency of 10 s. The baseline latency was first calculated from the average of 3 treatment-free latencies. The percentage maximum possible effect (% MPE) was then calculated for each test as follows:$$\%\;MPE=\frac{\left(test\;latency-baseline\;latency\right)}{\left(maximum\;latency-baseline\;latency\right)}$$

Cumulative dose effect responses were determined using a within-subject design. Here, mice were administered U50,488, **1**, **2**, or **3**, at increasing concentrations to create cumulative doses of 0.5, 1.0, 2.5, 5, 7.5, 10, 20, and 30 mg/kg (U50,488) or 0.05, 0.1, 0.3, 0.6, 1.0, 2.5, 5, and 7.5 mg/kg (analogues). Thirty minutes after each drug administration, the tail-withdrawal latency was measured, before the next cumulative dose was administered. The ED_50_, logED_50_, and E_max_ for each compound was determined by fitting a Log(agonist) vs. response, four parameter, non-linear-regression curve using GraphPad Prism (version 9.5.1). For duration of effect tests, C57BL/6J or KOR knockout mice were administered ED_50_ doses of U50,488 (7.6 mg/kg), **1** (0.6 mg/kg), **2** (0.9 mg/kg), **3** (0.55 mg/kg), or vehicle and tail withdrawal latencies were determined 5, 10, 15, 30, 45, 60, 90, 120, 150, and 180 min later.

### 4.8. Paclitaxel-Induced Allodynia

Chronic, paclitaxel-induced neuropathic pain was assessed as previously described [19]. During the induction phase (day 0–15), paclitaxel (0 or 4 mg/kg) or vehicle was administered on four alternate days (days 0, 2, 4, and 6). Mechanical and thermal allodynia were assessed on each of these days (prior to paclitaxel administration) and every 2–3 days thereafter to measure the progression of paclitaxel-induced neuropathic pain. These tests were conducted in elevated clear plastic chambers with a metal mesh floor. Mechanical allodynia was measured using an electric von Frey apparatus (Anesthesiometer 2390 series, IITC Life Science, Woodland Hills, CA, USA). The pressure filament (number 7) was applied at a right angle to the plantar surface of the hind paw and the force of pressure exerted to elicit a positive withdrawal response was recorded. Thermal allodynia was tested by applying a droplet of acetone to the plantar surface of the hind paw. The time taken to cause a positive reaction (elevating, biting, licking, or shaking of the paw) was measured, with 5 min between consecutive applications of acetone to the same paw. All tests were conducted by an experimenter blinded to the treatment conditions. For both types of stimulation, measurements were taken from each hind paw in duplicate, and averaged for each mouse.

During the treatment phase (day 18–42), mice that had received paclitaxel were then allocated to receive daily treatment of either U50,488 (7.0 mg/kg), **1** (0.6 mg/kg), **2** (1.1 mg/kg), **3** (0.55 mg/kg), or vehicle in a way that ensured an equivalent average mechanical allodynia score across the groups. These doses were based on the ID_80_ values calculated from a thermal allodynia pilot, which were similar to ED_50_ values in the warm water tail-withdrawal assay. Healthy control animals that received paclitaxel vehicle during the induction phase received KOR agonist vehicle during the treatment phase. Mechanical and thermal allodynia tests continued every 2–3 days.

### 4.9. Open-Field Locomotor Activity

Mice (C57BL/6J or KOR knockout) were administered U50,488 (7.6 mg/kg), **1** (0.6 mg/kg), **2** (1.1 mg/kg), **3** (0.55 mg/kg), or vehicle and immediately placed into the center of clean open-field chambers measuring 44 (width) × 44 (length) × 38 (height) cm. The distance traveled was measured in 5 min bins for 60 min thereafter using video tracking software (SMART software version 3.0, Panlab Harvard Apparatus). This experiment was replicated in C57BL/6J or β-arrestin2 knockout mice with U50,488 (10 mg/kg), **3** (0.5 or 1.0 mg/kg), or vehicle administration. A third replication was carried out with U50,488 (10 mg/kg), **3** (0.5 or 1.0 mg/kg), or vehicle being administered 15 min prior to testing, and the additional measurement of the number of entries and the time spent in the central 25 × 25 cm area during the first 10 min.

### 4.10. Passive Wire Hang

To measure fatigue and grip strength, we used a passive wire hang test [73,74]. Mice were placed on a wire frame, which was then slowly inverted and suspended 40 cm above a cushioned chamber. The latency to fall was measured in triplicate (with a 30 s interval in between trials) and the maximum cut-off latency was 75 s. Baseline measurements first taken, before U50,488 (7.6 mg/kg), **1** (0.6 mg/kg), **2** (1.1 mg/kg), **3** (0.55 mg/kg), or vehicle was administered, and measurements were taken at 30 min and 60 min thereafter. The average latency at baseline and at each time point was determined for each mouse, and the data were expressed a percentage change from baseline. The experiment was repeated as above but with U50,488 (10 mg/kg), **3** (0.5 or 1.0 mg/kg), or vehicle.

### 4.11. Rotarod

A four-lane accelerating rotarod (Harvard apparatus, Holliston, MA, USA) was set to accelerate from 3 to 30 rpm over 300 s. Mice (C57BL/6J or β-arrestin2 knockout mice) received 4 training sessions per day, for 5 consecutive days prior to testing. On the day of testing, the latency to fall was automatically measured at baseline (average of three) and at 15, 30, 45, 60, 90, and 120 min following the administration of U50,488 (10 mg/kg), **3** (0.5 or 1.0 mg/kg), or vehicle. Data were expressed as a percentage change from baseline.

### 4.12. Elevated Zero Maze

Mice were administered U50,488 (10 mg/kg), **3** (0.5 or 1.0 mg/kg), or vehicle 15 min prior to being placed into the closed arm, facing the open arm, of an elevated zero maze. The plastic maze had an annular gray track (5 cm wide, 40 cm internal diameter) that was separated into four quadrants, with two opposite quadrants being enclosed by 16 cm high dark, opaque walls, and two are open with 0.2 cm walls. The total distance traveled, open arm entries, and the time spent in the open arms of the maze were quantified using video tracking software (SMART v3.0 software, Panlab Harvard Apparatus, MA, USA).

### 4.13. Conditioned Place Aversion

A 4-day biased conditioned place aversion (CPA) procedure was carried out in a 3-chamber CPA apparatus based on previous methods [75]. The small central chamber was gray (size) and flanked by two larger chambers (size) with closable doors permitting movement between then. One of the large chambers contained a smooth black floor and black and white polka dot walls, whereas the other large chamber contained a white textured floor with black and white stripped walls. The time spent in each chamber was recorded using video tracking software (SMART v3.0 software, Panlab Harvard Apparatus).

On day 1 (pre-test), mice were allowed to freely roam the CPA chambers for 20 min. The percentage of time spent in each of the large chambers was recorded to determine the initial chamber preference for each mouse. On days 2–3 (conditioning), in the morning, mice received vehicle treatment 15 min prior to being confined to their non-preferred chamber for 40 min. In the afternoon (four hours later), mice received U50,488 (10 mg/kg), **3** (0.5 or 1.0 mg/kg), or vehicle 15 min prior to being confined to their preferred chamber for 40 min. On day 4 (post-test), mice were again allowed to freely roam the CPA chambers for 20 min, and the percentage preference for each chamber was determined.

### 4.14. Whole Body Plethysmography

Respiratory function was measured using whole body plethysmography based on previous methods [76,77], with two weeks of habituation carried out prior to testing. Briefly, a sealed plexiglass chamber was connected to an identical reference chamber via a differential pressure transducer, which was connected to a bridge amplifier (PowerLab 26T, AD Instruments, Dunedin, New Zealand) and recordings were charted using LabChart v8 software (AD Instruments). To calibrate the pressure recordings, 200 µL of air was injected into the chamber. The chamber was kept at 30 °C and a constant humidity by passing carbogen gas (5% O_2_ in CO_2_, BOC, Wellington, New Zealand) through a scintillated glass bead humidifier in a 75 °C water bath.

Male and female mice (C57BL6J or KOR knockout) were placed into the recording chamber for 10 min, during which baseline measurements were taken at 3, 7, and 9 min. Mice were then administered U50,488 (5.0 mg/kg), **1** (0.3 mg/kg), **3** (0.3 mg/kg), or vehicle and returned to the chamber. Recordings were taken at 5–10 min intervals for 60 min thereafter. Respiratory data from 5 s of clean trace (when the mice were not moving) at each time interval were analyzed by an experimenter blinded to the treatment conditions to calculate the tidal volume and respiratory frequency, which was expressed as a percentage change from baseline.

### 4.15. Statistical Analysis

Inferential statistics were carried out using one- or two-way analysis of variance (ANOVA). Significant effects from one-way ANOVA’s were followed up by Dunnett’s comparisons relative to control. Significant interaction effects from two-way ANOVA’s were followed up by simple Šídák’s (2 groups) or Dunnett’s (3+ groups) comparisons relative to control. Greenhouse–Geisser corrections were used for all repeated measures effects. Analyses were conducted using GraphPad Prism (version 9.5.1). All tests were two-sided, and results were considered statistically significant when *p* < 0.05. A summary of all statistical analyses can be found in the Appendix A.

## Data Availability

Data are presented within the manuscript. Additional raw data are available on request from the corresponding author.

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
