# Peer review of "Effects of Biased Analogues of the Kappa Opioid Receptor Agonist, U50,488, in Preclinical Models of Pain and Side Effects"

_molecules, 2025, doi:10.3390/molecules30030604_

Round 1
Reviewer 1 Report
Comments and Suggestions for Authors
This is a review of the manuscript “Effects of biased analogues of the kappa opioid receptor agonist, U50,488, in preclinical models of pain and side effects,” submitted by Ross van de Wetering , Loan Y Vu , Lindsay D Kornberger , Dan Luo , Brittany Scouller , Sheein Hong , Kelly Paton , Thomas E Prisinzano , Bronwyn M Kivell, submitted for publication in the journal Molecules. The manuscript contains data and findings of considerable interest to the fields of Pain Medication & Pharmacology and the Endogenous Opioid System regarding the effects of novel analoques of the well-studied κ opioid receptor agonist U50,488 in animal models of pain, as well as several animal models of known opioid side effects. There are however several areas the clarification of which would improve the quality and readability of the overall manuscript, as I detail below.
I was unfamiliar with the passive wire hang model of grip strength, and I found no evidence for it having been previously used in the context of KOR pharmacology; the closest reference was Sora et al 1997, although the application was different (MOR vs KOR) and terminology was also different. This is not a criticism of the use of the new model, but some commentary on it is warranted, as well as some references to guide those, like myself, who may be unfamiliar with interpreting the findings shown from this model.
The whole-body plethysmography data are derived from very low n studies, 2-3 for reqpiratory frequency. It was curious to see the statement in the method section that no statistical differences were observed in males versus females, and they were thus subsequently combined. It is difficult to judge the soundness of the conclusions that there were no effects of the analogues, in particular 1 which was trending lower in wildtype animals, given the low n.
The Introduction of the influence of G-protein bias on the side effect profile in KOR in the Introduction, lines 58-62, should probably be more nuanced – an improved side effect profile need not mean the elimination of all side effects. The findings from White et al in particular (58) supported G-protein bias resulting in reduction in sedation/incoordination side effects, if not in fact with respect to hedonic side effects.
The discrepancy in findings of beta-arrrestin-2 knockout mice in the White et al, where attenuation of the rotarod incoordination was indeed observed, compared to the findings here, where there was none, might warrant further comment as to possible reasons why the findings are divergent, or at the very least that future studies are warranted into the potential role of beta-arrestin-2 in KOR-induced motor incoordination.
References to the signaling bias model(s) used for the bias factor calculation should be given.
Author Response
Response to Reviewer 1 Comments
- Summary
Thank you very much for taking the time to review this manuscript. Please find our detailed responses below and the corresponding revisions to your comments.
- Questions for General Evaluation Reviewer's Evaluation Response and Revisions
Does the introduction provide sufficient background and include all relevant references? Yes/Can be improved/Must be improved/Not applicable See below
Is the research design appropriate? Yes/Can be improved/Must be improved/Not applicable
Are the methods adequately described? Yes/Can be improved/Must be improved/Not applicable
Are the results clearly presented? Yes/Can be improved/Must be improved/Not applicable
Are the conclusions supported by the results? Yes/Can be improved/Must be improved/Not applicable
- Point-by-point response to Comments and Suggestions for Authors
Comments 1: This is a review of the manuscript "Effects of biased analogues of the kappa opioid receptor agonist, U50,488, in preclinical models of pain and side effects," submitted by Ross van de Wetering , Loan Y Vu , Lindsay D Kornberger , Dan Luo , Brittany Scouller , Sheein Hong , Kelly Paton , Thomas E Prisinzano , Bronwyn M Kivell, submitted for publication in the journal Molecules. The manuscript contains data and findings of considerable interest to the fields of Pain Medication & Pharmacology and the Endogenous Opioid System regarding the effects of novel analoques of the well-studied κ opioid receptor agonist U50,488 in animal models of pain, as well as several animal models of known opioid side effects. There are however several areas the clarification of which would improve the quality and readability of the overall manuscript, as I detail below.
Response 1: Thank you for your constructive comments. We have addressed all the areas you mention, as detailed below.
Comments 2: I was unfamiliar with the passive wire hang model of grip strength, and I found no evidence for it having been previously used in the context of KOR pharmacology; the closest reference was Sora et al 1997, although the application was different (MOR vs KOR) and terminology was also different. This is not a criticism of the use of the new model, but some commentary on it is warranted, as well as some references to guide those, like myself, who may be unfamiliar with interpreting the findings shown from this model.
Response 2: The passive wire hang test can be used to measure sedation/fatigue and grip strength; it is more commonly used in neuro/muscular degeneration research.
We have cited a methods paper and another paper that have used this procedure in our methods.
- Section 4.10 - line 442: "To measure fatigue and grip strength, we used a passive wire hang test [73,74]."
In the results section, we have added further clarified what the results from the passive wire test indicate.
- Section 2.4 - line 167-170: "In the passive wire hang test, U50,488, 1, and 2 significantly decreased hang latency compared to vehicle treated controls, indicative of increased fatigue and/or reduced grip strength, whereas 3 had no significant effect (Figure 5a)."
Comments 3: The whole-body plethysmography data are derived from very low n studies, 2-3 for reqpiratory frequency. It was curious to see the statement in the method section that no statistical differences were observed in males versus females, and they were thus subsequently combined. It is difficult to judge the soundness of the conclusions that there were no effects of the analogues, in particular 1 which was trending lower in wildtype animals, given the low n.
Response 3: As mentioned in the manuscript, selective KOR agonists are not known to induce respiratory depression (refs 28, and 29), therefore, we would not expect to see a significant result here.
Nevertheless, we do definitely agree that the sample size of wild-type mice was low, and we have now mentioned this limitation in the manuscript.
- Section 2.4 - line 184-187: "Accordingly, we found that U50,488, 1, and 2 had no significant effect on respiratory frequency (Figure 7a-b) or tidal volume (Figure 7c-d) in wild-type or KOR knockout mice, though, we must acknowledge the low sample size of wild-type mice used in this experiment may limit statistical power."
We have also excluded the statement in the methods on the effects in males versus females, since we do not have the sample size to reliably conclude this in both wildtype and knockout mice.
Comments 4: The Introduction of the influence of G-protein bias on the side effect profile in KOR in the Introduction, lines 58-62, should probably be more nuanced - an improved side effect profile need not mean the elimination of all side effects. The findings from White et al in particular (58) supported G-protein bias resulting in reduction in sedation/incoordination side effects, if not in fact with respect to hedonic side effects.
Response 4: We've brought forward some of the material from the discussion to this introduction paragraph and provided additional details on this topic to respond to this comment and another reviewer's (on clarifying the role of beta arrestin 1 versus 2). This includes more clearly stating the hypothesis that increased G-protein bias may increase therapeutic windows / minimize side effects (not eliminate all side effects). The paragraph has now been split in two.
- Section 1 - Lines 51-70: "However, KOR agonists have their own adverse side effects, including sedation, anxiety, and dysphoria/aversion [31-41], which has limited their clinical development. In an effort to minimize these side effects, many researchers have sought to develop novel compounds that preferentially stimulate KOR G-protein signaling over β-arrestin [14-18]. This idea, at least within the KOR field, largely stems from research showing that blockade of p38 mitogen-activated protein kinase (MAPK) activation using the inhibitor, SB203580, or a conditional knockout of p38α MAPK, prevented conditioned place aversion caused by the prototypical KOR agonist, U50,488, in mice [32,42]. Activation of p38 MAPK is thought to be dependent on β-arrestin2 recruitment by KOR [43]. Therefore, it was hypothesized that reduced β-arrestin2 recruitment, which does not appear to impact the antinociceptive effects of KOR agonists [44-46], may increase the therapeutic windows [47].
While several novel G-protein biased KOR agonists have shown reduced side effects compared to typical KOR agonists [14-18], it is not clear whether the G-protein bias of these compounds is indeed the cause for their reduced side effects since some very highly G-protein biased KOR agonists still showed significant aversive or sedative effects [48,49]. Moreover, subsequent research has shown that knockout of β-arrestin2 had no impact on conditioned place aversion or locomotor sedation caused by a variety of KOR agonists, but it did attenuate KOR-induced motor incoordination on the rotarod [49]. These inconsistent findings question whether G-protein bias is a useful metric to pursue for KOR drug development."
We have mentioned in the introduction and discussion that there are several G-protein biased KOR agonists that show fewer side effects - this includes the study by White et al., evaluating RB-64 which did cause conditioned place aversion, but not locomotor sedation, motor incoordination, and had no effect on ICSS.
However, as we now more clearly outline in the discussion (see below comments), we believe this is not the strongest evidence for a causal relationship between G-protein bias and side effects. Therefore, we focus on their experiments with beta arrestin knockout mice, which largely show no effect except for an attenuation on motor incoordination. It would have been interesting to the effect of beta arrestin knockout in their ICSS studies.
Comments 5: The discrepancy in findings of beta-arrrestin-2 knockout mice in the White et al, where attenuation of the rotarod incoordination was indeed observed, compared to the findings here, where there was none, might warrant further comment as to possible reasons why the findings are divergent, or at the very least that future studies are warranted into the potential role of beta-arrestin-2 in KOR-induced motor incoordination.
Response 5: We have revised the discussion section on G-protein bias and beta arrestin signaling (section 3.2 - lines 272-310) to more thoroughly and clearly discuss the results of previous research and ours, including more detailed comments on this discrepant finding.
- Section 3.2 - line 302-310: "In regard to sedation, White and colleagues found that knockout of β-arrestin2 had no impact on locomotor sedation caused by U69,593 or salvinorin A [49], which is in line with the findings of the current study. They also found that both U69,593 and salvinorin A still caused significant motor incoordination on the rotarod in β-arrestin2 knockout mice, though this was attenuated compared to wild type controls [49]. These results suggest that β-arrestin2 signaling may contribute to motor incoordination, but it is not necessary. In the current study, we found that knockout of β-arrestin2 had no effect on U50,488-induced motor incoordination. The reason for this discrepancy is unclear, but it may be a drug-dependent effect."
Reviewer 2 Report
Comments and Suggestions for Authors
This manuscript presents the characterization of three new analogs of the kappa opioid receptor agonist U50,488. It describes the synthesis of the analogs, in vitro analysis of signaling bias, and antinociceptive effects and undesired side effects in vivo. The methods used were described well. The results were clearly presented, and appropriate statistical analyses were provided. The interpretation of the results is mostly agreeable. The use of the kappa opioid receptor knockout mice strengthens the conclusion. However, this reviewer cannot agree with the suggestion that β-arrestin2 had no role in mediating KOR-induced sedation or motor incoordination. β-arrestin1 and β-arrestin2 have high sequence homology and overlapped expression. Their functional compensation has been suggested. This possibility should be discussed. The rationale for choosing the β-arrestin2 knockout mouse over the β-arrestin1 knockout mouse should be presented. In the text, there are “β-arrestin” without “2”. Are these typos or intentional? If they are typos, they should be fixed. If intentional, the authors should discuss the potential role of β-arrestin1. The discovery/synthesis and characterization of a promising lead compound (3) with therapeutic potential is significant and supports the publication of this manuscript after revision.
Author Response
|
Response to Reviewer 2 Comments
|
||
|
1. Summary |
|
|
|
Thank you for reviewing our manuscript submission. We have revised our manuscript based on your comments and believe it has been improved as a result.
|
||
|
2. Questions for General Evaluation |
Reviewer’s Evaluation |
Response and Revisions |
|
Does the introduction provide sufficient background and include all relevant references? |
Yes/Can be improved/Must be improved/Not applicable |
See below |
|
Is the research design appropriate? |
Yes/Can be improved/Must be improved/Not applicable |
|
|
Are the methods adequately described? |
Yes/Can be improved/Must be improved/Not applicable |
|
|
Are the results clearly presented? |
Yes/Can be improved/Must be improved/Not applicable |
|
|
Are the conclusions supported by the results? |
Yes/Can be improved/Must be improved/Not applicable |
|
|
3. Point-by-point response to Comments and Suggestions for Authors |
||
|
Comments 1: This manuscript presents the characterization of three new analogs of the kappa opioid receptor agonist U50,488. It describes the synthesis of the analogs, in vitro analysis of signaling bias, and antinociceptive effects and undesired side effects in vivo. The methods used were described well. The results were clearly presented, and appropriate statistical analyses were provided. The interpretation of the results is mostly agreeable. The use of the kappa opioid receptor knockout mice strengthens the conclusion. However, this reviewer cannot agree with the suggestion that β-arrestin2 had no role in mediating KOR-induced sedation or motor incoordination. β-arrestin1 and β-arrestin2 have high sequence homology and overlapped expression. Their functional compensation has been suggested. This possibility should be discussed. The rationale for choosing the β-arrestin2 knockout mouse over the β-arrestin1 knockout mouse should be presented. In the text, there are “β-arrestin” without “2”. Are these typos or intentional? If they are typos, they should be fixed. If intentional, the authors should discuss the potential role of β-arrestin1. The discovery/synthesis and characterization of a promising lead compound (3) with therapeutic potential is significant and supports the publication of this manuscript after revision.
|
||
|
Response 1: We have revised our discussion on the role of G-protein bias / beta arrestin signaling to more clearly discuss the results of our research in the context of previous findings, to better justify our conclusions on the role of beta arrestin2 and KOR side effects.
- Section 3.2 – lines 272-313: “In the search for therapeutically viable KOR agonists with fewer adverse side effects, many research groups, including ours, have aimed to develop compounds with G-protein signaling bias. Although several novel G-protein biased KOR agonists have shown im-proved side effects profiles compared to typical KOR agonists [14–18], it is difficult to attribute any reduction in side effects to their G-protein bias for two primary reasons. Firstly, these findings are largely correlational, and do not imply a causal role of G-protein bias in minimizing adverse side effects. Moreover, there are also several contradictory findings, with some highly G-protein biased compounds still showing significant side effects. For example, the salvinorin A derivative, RB-64, caused similar significant conditioned place aversion as salvinorin A at equivalent antinociceptive doses, despite being 96 times more bias for G-protein signaling compared to salvinorin A [49]. In another example with diphenethylamine derivatives, HS665 (with a G-protein bias factor of 389 compared to U69,539) showed significant conditioned place aversion while HS666 (with a G-protein bias factor of 62 compared to U69,539) did not [48]. In our study, 1 (bias factor 0.9) and 2 (bias factor 2.2) showed significant sedative effects while 3 (bias factor 1.8) did not. To reliably determine if G-protein bias correlates with greater therapeutic windows, a systematic investigation using several KOR agonists with different G-protein bias is re-quired, as has been explored with some MOR agonists [57]. Furthermore, it would be beneficial to test β-arrestin biased compounds to determine whether the opposite holds true, i.e., do β-arrestin biased KOR agonists show more side effects? Second, studies that have directly investigated the causal role of G-protein bi-as/β-arrestin2 signaling have shown conflicting results. Inhibition of β-arrestin2-dependant p38 MAPK signaling prevented conditioned place aversion to U50,488, suggesting a role of β-arrestin2 and p38 MAPK in KOR-induced aversion [32,42]. In contrast, β-arrestin2 knockout mice still showed significant conditioned place aversion to the KOR agonists, U69,593, salvinorin A, and RB-64, with no difference compared to wild-type mice [49]. The authors suggested that KOR-induced aversion could instead be mediated by G-protein signaling, by p38-MAPK-independent mechanisms, or that p38-MAPK could be activated β-arrestin2-independent mechanisms [49]. Given the global knockout model used in their study, as well as in the current study, it is possible that β-arrestin1 may compensate as the primary mediator of p38 MAPK signaling. In regard to sedation, White and colleagues found that knockout of β-arrestin2 had no impact on locomotor sedation caused by U69,593 or salvinorin A [49], which is in line with the findings of the current study. They also found that both U69,593 and salvinorin A still caused significant motor incoordination on the rotarod in β-arrestin2 knockout mice, though this was attenuated compared to wild type controls [49]. These results suggest that β-arrestin2 signaling may contribute to motor incoordination, but it is not necessary. In the current study, we found that knockout of β-arrestin2 had no effect on U50,488-induced motor in-coordination. The reason for this discrepancy is unclear, but it may be a drug-dependent effect. Taken together, current evidence does not support a clear role of β-arrestin signaling in KOR agonist-induced side effects and G-protein bias does to appear to be a reliable pre-dictor of therapeutic viability….”
We have also revised our introductory paragraphs to clarify why we (and others) have focused on beta arrestin2 specifically.
- Section 1 – lines 51-61: “However, KOR agonists have their own adverse side effects, including sedation, anxiety, and dysphoria/aversion [31–41], which has limited their clinical development. In an effort to minimize these side effects, many researchers have sought to develop novel compounds that preferentially stimulate KOR G-protein signaling over β-arrestin [14–18]. This idea, at least within the KOR field, largely stems from research showing that blockade of p38 mitogen-activated protein kinase (MAPK) activation using the inhibitor, SB203580, or a conditional knockout of p38α MAPK, prevented conditioned place aversion caused by the prototypical KOR agonist, U50,488, in mice [32,42]. Activation of p38 MAPK is thought to be dependent on β-arrestin2 recruitment by KOR [43]. Therefore, it was hypothesized that reduced β-arrestin2 recruitment, which does not appear to impact the antinociceptive effects of KOR agonists [44–46], may increase the therapeutic windows [47].”
Furthermore, in the discussion, we had mentioned the possibility of functional compensation by beta arrestin1 when interpreting findings from beta arrestin KO mice
- Section 3.2 – lines 300-302: “Given the global knockout model used in their study, as well as in the current study, it is possible that β-arrestin1 may compensate as the primary mediator of p38 MAPK signaling.”
|
||
Reviewer 3 Report
Comments and Suggestions for Authors
In this MSS entitled 'Effects of biased analogues of the kappa opioid receptor agonist, U50,488, in preclinical models of pain and side effects" Ross van de Wetering et al., presented characterization of three compounds for acute and chronic antinociceptive effects. Authors look for behavioral studies in addition to determine the levels of second messenger cAMP.
In general this MSS is well written and presented MSS except few sentences and authors are advised to read the MSS carefully.
Describe results for compound U50,488 first with all details . It is necessary because rest of the three compounds 1, 2 and 3 are compared with this main drug. Discuss briefly about the compound U50-488.
In Figure 1, please add name of the precursor compound used for the synthesis of 1, 2 and 3.
Beta arrestin is a critical proteins in structure function and regulation of GPCRs but has not been discussed here to establish a possible link with the present study. As discussed beta arrestin KO play no significant role in characterization of these compounds.
Some compound have no side effect, these are fascinating observation and need to be discussed mechanistically which is essential for therapeutic implication.
Material and Method section is well described.
If I am correct authors did not use any MOR agonist here, this study would have been very complete if MOR agonist were used for comparison but discussed briefly in line 240-242.
Second important component missing some downstream signaling pathways as discussed in line 255-257.
These two additional experiments will definitely strengthen this presentation but not a urgent requirement.
Author Response
|
Response to Reviewer 3 Comments
|
||
|
1. Summary |
|
|
|
We appreciate your comments. Please see our detailed responses below.
|
||
|
2. Questions for General Evaluation |
Reviewer’s Evaluation |
Response and Revisions |
|
Does the introduction provide sufficient background and include all relevant references? |
Yes/Can be improved/Must be improved/Not applicable |
See below |
|
Is the research design appropriate? |
Yes/Can be improved/Must be improved/Not applicable |
|
|
Are the methods adequately described? |
Yes/Can be improved/Must be improved/Not applicable |
|
|
Are the results clearly presented? |
Yes/Can be improved/Must be improved/Not applicable |
|
|
Are the conclusions supported by the results? |
Yes/Can be improved/Must be improved/Not applicable |
|
|
3. Point-by-point response to Comments and Suggestions for Authors |
||
|
Comments 1: In this MSS entitled 'Effects of biased analogues of the kappa opioid receptor agonist, U50,488, in preclinical models of pain and side effects" Ross van de Wetering et al., presented characterization of three compounds for acute and chronic antinociceptive effects. Authors look for behavioral studies in addition to determine the levels of second messenger cAMP.
In general this MSS is well written and presented MSS except few sentences and authors are advised to read the MSS carefully.
|
||
|
Response 1: Thank you for your comments, these have all been addressed, as detailed below.
|
||
|
Comments 2: Describe results for compound U50,488 first with all details . It is necessary because rest of the three compounds 1, 2 and 3 are compared with this main drug. Discuss briefly about the compound U50-488.
|
||
|
Response 2: Where possible we have made revisions in order to describe the results of U50,488 before the other compounds. For example:
- Section 2.4 – lines 156-158: “Knockout of β-arrestin2 had no impact of U50,488-induced sedation, while 3 had no significant sedative effect at either dose (Figure 4c, d).” - Section 2.4 – lines 158-164: “When tested with a 15 min pre-treatment period to evaluate novelty/anxiety-induced exploration, U50,488 significantly decreased the number of entries into the central zone, suggesting an anxiogenic like-effect (Figure 4f), though this could also be due to the sedative effects of the compound, since there was no effect on the percent time spent in the center (Figure 4g). In contrast, 3 continued to show no sedative effects, even at 2×ED50 doses (Figure 4e), and also had no effect on the number of entries (Figure 4f) or the percent time spent (Figure 4g) in the center of the open field arena, indicating no impact on anxiety-like thigmotaxis behavior.”
|
||
|
Comments 3: In Figure 1, please add name of the precursor compound used for the synthesis of 1, 2 and 3.
|
||
|
Response 3: This compound is mentioned in the introduction. We have now labelled this compound as compound A in text, in Figure 1, and in the Figure 1 caption.
- Section 1 – line 72-74: “Previously, Weerawarna and colleagues found that N-methyl-[(1S)-phenyl-2-(l-pyrrolidinyl)ethyl]ethyl]phenylacetamide (compound A; Figure 1) had high affinity for the KOR [50].” - Figure 1 caption – lines 89-90: “Figure 1. Chemical structures of 1, 2, and 3, which were inspired by N-methyl-[(1S)-phenyl-2-(l-pyrrolidinyl)ethyl]ethyl]phenylacetamide (compound A) [50].”
|
||
|
Comments 4: Beta arrestin is a critical proteins in structure function and regulation of GPCRs but has not been discussed here to establish a possible link with the present study. As discussed beta arrestin KO play no significant role in characterization of these compounds.
|
||
|
Response 4: Our revised introduction more clearly discusses the role of beta arrestin 2 signaling and how it can impact downstream signaling targets such as p38 that might mediate aversion. - Section 1 – lines 52-61: “In an effort to minimize these side effects, many researchers have sought to develop novel compounds that preferentially stimulate KOR G-protein signaling over β-arrestin [14–18]. This idea, at least within the KOR field, largely stems from research showing that blockade of p38 mitogen-activated protein kinase (MAPK) activation using the inhibitor, SB203580, or a conditional knockout of p38α MAPK, prevented conditioned place aversion caused by the prototypical KOR agonist, U50,488, in mice [32,42]. Activation of p38 MAPK is thought to be dependent on β-arrestin2 recruitment by KOR [43]. Therefore, it was hypothesized that reduced β-arrestin2 recruitment, which does not appear to impact the antinociceptive effects of KOR agonists [44–46], may increase the therapeutic windows [47].”
|
||
|
Comments 5: Some compound have no side effect, these are fascinating observation and need to be discussed mechanistically which is essential for therapeutic implication.
|
||
|
Response 5: We have added a paragraph on this.
- Section 3.1 – lines 261-269: “One possible explanation for the unique side effect profile of 3 could be due to differences in logD value (3: LogD = 1.9, 1: LogD = 2.7, 2: LogD = 2.6). The replacement of a phenyl ring to a pyridine 3 results in a less lipophilic compound which could impact the ligand-protein affinity [63]. Daibani and colleagues reported in their molecular dynamic simulation that U50,488 inserted a hydrophobic group, which was the cyclohexane ring in the central pocket [64]. If our analogues possessed a similar binding mode, the interaction of pyridine 3 to the central pocket would result in a unique binding mode compared to phenyl ring present in 1 and 2. How this might impact downstream cellular signaling and side effects is not entirely clear, but as discussed below, we do not think β-arrestin2 signaling plays an important role.”
|
||
|
Comments 6: Material and Method section is well described.
|
||
|
Response 6: Thanks
|
||
|
Comments 7: If I am correct authors did not use any MOR agonist here, this study would have been very complete if MOR agonist were used for comparison but discussed briefly in line 240-242.
|
||
|
Response 7: That is correct, we did not use any MOR agonists here, this research focused on KOR agonists. These experiments have been previously conducted using MOR agonists, by us and others, and we did not think it necessary to repeat those here.
We have already discussed our findings with respect to MOR agonists. For example: - Section 3.1 – lines 230-234: “While these compounds were all ~10× more potent than the prototypical KOR agonist, U50,488, they were not more efficacious, and the maximum efficacy (~50%MPE) was notably lower than that achieved with MOR agonists (100%MPE) [57]. This may limit their therapeutic potential for managing severe acute pain.” - Section 3.1 0 lines 239 – 244: “MOR agonists, while highly effective for acute severe pain, have significant on-target side effects that limit their safety and utility in managing such chronic pain [60]. Our results suggest that KOR agonists may provide a safer alternative. We found that all KOR agonists completely reversed paclitaxel-induced allodynia in mice. Importantly, unlike MOR agonists [19], we observed no evidence of tolerance, even after 3 weeks of daily treatment.”
We believe this is sufficient and that further discussion on the effects of MOR agonists or the role of G-protein bias in the effects of MOR agonists is outside of the scope of the current research. |
||
|
Comments 8: Second important component missing some downstream signaling pathways as discussed in line 255-257.
|
||
|
Response 8: See our response to comment 4 regarding the new reference to downstream signaling pathways.
|
||
|
Comments 9: These two additional experiments will definitely strengthen this presentation but not a urgent requirement. |
||
|
Response 9: We believe we have addressed all of the above comments with references to the available literature.
|
||
Round 2
Reviewer 2 Report
Comments and Suggestions for Authors
This reviewer is satisfied with the authors' response and the revised manuscript.